# Combined *PIK3CA* and *SOX2* Gene Amplification Predicts Laryngeal Cancer Risk beyond Histopathological Grading

**DOI:** 10.3390/ijms25052695

**Published:** 2024-02-26

**Authors:** Irene Montoro-Jiménez, Rocío Granda-Díaz, Sofía T. Menéndez, Llara Prieto-Fernández, María Otero-Rosales, Miguel Álvarez-González, Vanessa García-de-la-Fuente, Aida Rodríguez, Juan P. Rodrigo, Saúl Álvarez-Teijeiro, Juana M. García-Pedrero, Francisco Hermida-Prado

**Affiliations:** 1Department of Otolaryngology, Hospital Universitario Central de Asturias and Instituto de Investigación Sanitaria del Principado de Asturias (ISPA), Instituto Universitario de Oncología del Principado de Asturias (IUOPA), University of Oviedo, 33011 Oviedo, Spain; imj21897@gmail.com (I.M.-J.); rocigd281@gmail.com (R.G.-D.); sofiatirados@gmail.com (S.T.M.); prietollara@uniovi.es (L.P.-F.); uo290605@uniovi.es (M.O.-R.); miguelag.2296@gmail.com (M.Á.-G.); vgarcifuente@gmail.com (V.G.-d.-l.-F.); aida.rodriguez@ispasturias.es (A.R.); jprodrigo@uniovi.es (J.P.R.); saul.teijeiro@gmail.com (S.Á.-T.); franjhermida@gmail.com (F.H.-P.); 2CIBERONC, Instituto de Salud Carlos III, 28029 Madrid, Spain

**Keywords:** *SOX2*, *PIK3CA*, 3q26 gene amplification, laryngeal tumorigenesis, cancer risk marker, dysplasia

## Abstract

The *PIK3CA* and *SOX2* genes map at 3q26, a chromosomal region frequently amplified in head and neck cancers, which is associated with poor prognosis. This study explores the clinical significance of *PIK3CA* and *SOX2* gene amplification in early tumorigenesis. Gene copy number was analyzed by real-time PCR in 62 laryngeal precancerous lesions and correlated with histopathological grading and laryngeal cancer risk. Amplification of the *SOX2* and *PIK3CA* genes was frequently detected in 19 (31%) and 32 (52%) laryngeal dysplasias, respectively, and co-amplification in 18 (29%) cases. The *PIK3CA* and *SOX2* amplifications were predominant in high-grade dysplasias and significantly associated with laryngeal cancer risk beyond histological criteria. Multivariable Cox analysis further revealed *PIK3CA* gene amplification as an independent predictor of laryngeal cancer development. Interestingly, combined *PIK3CA* and *SOX2* amplification allowed us to distinguish three cancer risk subgroups, and *PIK3CA* and *SOX2* co-amplification was found the strongest predictor by ROC analysis. Our data demonstrate the clinical relevance of *PIK3CA* and *SOX2* amplification in early laryngeal tumorigenesis. Remarkably, *PIK3CA* amplification was found to be an independent cancer predictor. Furthermore, combined *PIK3CA* and *SOX2* amplification is emerging as a valuable and easy-to-implement tool for cancer risk assessment in patients with laryngeal precancerous lesions beyond current WHO histological grading.

## 1. Introduction

Head and neck squamous cell carcinoma (HNSCC) is the sixth most common malignancy worldwide, accounting for 890,000 new cases and approximately 450,000 deaths each year [1]. This tumor type arises in the mucosal surfaces of the upper aerodigestive tract, primarily located in the larynx, pharynx, and oral cavity. Notably, laryngeal carcinomas represent one-third of all head and neck cancers [2]. Tobacco and alcohol consumption and oncogenic viruses such as human papillomavirus (HPV) are recognized as major etiological factors [3]. Tobacco and alcohol consumption shows a synergistic effect on HNSCC development. Beyond these two etiological factors, HPV-related cancers have been continuously increasing, particularly in developed countries. Nowadays, HPV infection is responsible not only for most cancers that originate in the oropharynx, but also a number of laryngeal carcinomas [4,5]. Five-year survival rates for early stage I and II laryngeal carcinoma range from 80 to 90%, whereas survival markedly decreases to 50–60% for advanced stage III and IV tumors [6,7,8]. Patient mortality is mainly due to the late diagnosis and failure of loco-regional control, thus highlighting the major clinical need for novel methods of cancer detection and prognostication.

Laryngeal carcinomas, like other HNSCCs, encompass highly complex heterogeneous molecular alterations [7,9]. In particular, human papillomavirus (HPV)-negative tumors harbor many different genetic aberrations caused by carcinogen exposure such as TP53 loss-of-function, *CDKN2A* inactivation, and amplification of the 11q13 and 3q26-27 chromosomal regions [10,11].Specifically, 3q26-27 amplification is one of the most recurrent genetic alterations in squamous cell carcinomas of the larynx and other anatomical sites, and has been associated with tumor progression and poor disease outcome [12,13,14,15,16].

*PIK3CA* and *SOX2* genes have emerged as major oncogenic drivers within the 3q26 amplicon [17]. *PIK3CA* encodes the p110α catalytic subunit of phosphatidylinositol 3-kinase (PI3K), which is among the most commonly mutated and amplified genes in HNSCC and other cancers, and PI3K signaling is the most frequently altered oncogenic pathway [18,19,20]. PI3K signaling is implicated in various key cellular and biological processes that contribute to tumor progression including the regulation of metabolism, cell growth, proliferation, metastasis, and resistance to therapy [21,22,23,24]. SOX2 is a member of the SOX (SRY-related HMG-box) family of transcription factors and a critical player in embryonic development, stem cell maintenance, and oncogenesis [25,26,27]. SOX2 expression is commonly detected in multiple cancer types and found to positively regulate tumor cell proliferation, migration, invasion, and metastasis [28].

Great effort has been devoted to identifying novel diagnostic or prognostic markers that reliably discriminate tumor behavior to improve patient stratification and prediction of outcome beyond the current clinical and histopathological criteria. In this regard, a new WHO classification has recently been established for laryngeal dysplasia into low-grade versus high-grade dysplasia, which attempts to overcome the limited predictability of previous three-tier grading as mild, moderate, and severe dysplasia [29,30]. Improving the laryngeal cancer diagnosis continues to be a priority research area aimed at detecting cancer at an earlier and more curable stage, which nowadays remains an unmet need. On this basis, the present study investigated the clinical significance of *PIK3CA* and *SOX2* gene amplification in early stages of laryngeal tumorigenesis by evaluating their predictive value for cancer risk assessment in 62 patients with laryngeal precancerous lesions.

## 2. Results

### 2.1. PIK3CA and SOX2 Gene Copy Amplification in Early Stages of Laryngeal Tumorigenesis

Amplification of the *SOX2* and *PIK3CA* genes was frequently detected in 19 (31%) and 32 (52%) out of 62 laryngeal dysplasias, respectively. Co-amplification of both genes was present in 18 (29%) cases (Table 1). The relative copy numbers ranged from 2- to 6-fold for *PIK3CA* amplification, and from 2- to 9-fold for *SOX2* amplification. In addition, a strong positive correlation was observed between amplification of the *PIK3CA* and *SOX2* genes (Figure 1; *p* < 0.001; Spearman coefficient = 0.450). Interestingly, gene amplification was found to gradually increase along the early stages of laryngeal tumorigenesis, as shown in Figure 2; however, *PIK3CA* amplification occurred at a higher frequency, reaching over 50% of cases with severe dysplasia. We also performed immunohistochemical analysis to confirm the presence of both proteins in various cases harboring amplification of the *PIK3CA* and *SOX2* genes (Appendix A).

*PIK3CA* and *SOX2* amplifications increased with the grade of dysplasia, and both were predominantly detected in high-grade dysplasias. *SOX2* gene amplification was observed in 2/8 (25%) low-grade dysplasias and 17/54 (31%) high-grade dysplasias. In turn, *PIK3CA* gene amplification was detected at a higher frequency in 3/8 (37.5%) low-grade dysplasias and 29/54 (54%) high-grade dysplasias. However, the differences did not reach statistical significance (*p* = 1.00 and *p* = 0.467, respectively; Fisher’s exact test).

### 2.2. Associations with Laryngeal Cancer Risk

During the follow-up period (mean 6.7 months, range 8 to 120 months), 24 (39%) out of 62 patients developed an invasive carcinoma at the biopsy site. The mean time to cancer diagnosis in the cases that progressed was 28.7 months (range 11 to 66 months). No significant differences attributable to age were observed (*p* = 0.870) between the group of patients who developed cancer (mean, 64.57 years) and those who did not (mean, 65.05 years).

Table 2 summarizes the associations of different parameters with the evolution of the lesions to develop an invasive carcinoma, and Figure 3 shows Kaplan–Meier curves for cancer-free survival. Age, tobacco, and histology were not found to significantly correlate with the risk of progression to laryngeal cancer (Table 2, Figure 3A,B, and Appendix A). Older patients (>70 years) in this study cohort showed a lower progression risk, although differences did not reach statistical significance (Table 2 and Appendix A). In contrast, lesions harboring *PIK3CA* gene amplification exhibited a significantly higher laryngeal cancer risk than negative lesions (Figure 3C, Log-rank test, *p* = 0.019; HR = 2.74, 95% CI 1.13–6.63, Cox regression *p* = 0.025). A similar trend was observed for *SOX2* amplification (Figure 3D, Log-rank test *p* = 0.031; HR = 2.36, 95% CI 1.05–5.29, Cox regression *p* = 0.037). Multivariable Cox analysis including age (cut-off = 70 years), tobacco consumption, WHO histological classification, *PIK3CA* amplification, and *SOX2* amplification showed that *PIK3CA* amplification was the only significant independent predictor of laryngeal cancer development (HR = 2.64, 95% CI 1.09–6.37, *p* = 0.031).

The combination of *PIK3CA* and *SOX2* amplification also significantly predicted the laryngeal cancer risk (Figure 3E, Log-rank test *p* = 0.033). Interestingly, patients with negative gene amplification showed the lowest risk of developing laryngeal cancer, whereas patients carrying lesions with amplification of both genes exhibited the highest risk (HR = 3.31, 95% CI 1.27–8.58, Cox regression *p* = 0.014), and patients with amplification of a single gene showed an intermediate cancer risk (HR = 1.76, 95% CI 0.59–5.25, Cox regression *p* = 0.31). Furthermore, combined amplification of both the *PIK3CA* and *SOX2* genes was found the strongest predictor of laryngeal cancer by ROC curve analysis (Table 3, AUC = 0.675, *p* = 0.021) (Figure 3F).

## 3. Discussion

Survival rates for HNSCC patients have not significantly changed over the last several decades. Mortality remains high, mainly due to a late cancer diagnosis and frequent recurrences and metastasis. Hence, early cancer biomarkers and accurate risk stratification methods represent crucial tools in the clinical practice to improve HNSCC diagnosis and develop prevention strategies.

HNSCC carcinogenesis has been defined as a multistep process that evolves by accumulation of morphological and molecular changes in the epithelial cells caused by carcinogen exposure. This results in premalignant lesions transforming into invasive carcinoma [31]. Histopathological diagnosis of squamous intraepithelial lesions is considered the ‘gold standard’ for cancer risk evaluation in clinical routine [32]. The presence of lesions with dysplastic features is related to a higher cancer risk; however, some cancers develop from lesions lacking dysplastic changes. A new WHO classification has recently been established for laryngeal dysplasia into low-grade versus high-grade dysplasia [29,30], which attempts to overcome the limited predictability of the previous 3-tier grading system. None of these classifications significantly predicted laryngeal cancer risk in our cohort, thus reflecting the still limited value of histologic grading in accurately predicting outcome. This result emphasizes the need for additional objective and reliable markers to improve cancer diagnosis, risk stratification, and treatment decision-making.

HNSCC development encompasses a number of genetic and genomic alterations that may offer an opportunity to identify powerful early biomarkers that complement histological evaluation for cancer risk assessment. In this sense, copy number variations (CNVs) are frequent genetic abnormalities found in many human diseases [33]. Copy number gains at chromosome 3q26 have emerged as one of the most common CNVs, which are detected in about 20% of all human cancers. The widespread presence and high frequency of 3q26 amplification suggest that this genetic alteration may play a relevant role in shaping tumor development and behavior [17]. Chromosome 3q harbors a large number of genes associated with unfavorable outcome [34], chemotherapy resistance [35], and reduced survival [36,37]. Amplification of 3q is frequently detected in early stages of tumorigenesis. This has also been considered a driver genetic event in lung cancer development and uterine carcinoma progression [38,39]. In the context of HNSCC, 3q amplification is common and related to tumor progression [40,41]. Notably, 3q amplification has been reported in 3% of normal mucosa, 25% of precancerous lesions, and 56% of invasive HNSCCs [15]. Bioinformatic analyses in lung squamous cell carcinomas have revealed 35 potential driver genes within the recurrent 3q26 amplicon. Among them, *PIK3CA, SOX2, ECT2,* and *PRKCI* have been highlighted as the most promising candidate oncogenes that could cooperatively operate in the intracellular signaling network [17].

The present study sheds light on the importance of *PIK3CA* and *SOX2* amplification in the early stages of laryngeal tumorigenesis. Both amplifications were detected in patients with laryngeal precancerous lesions, and gradually increased with the grade of dysplasia. Nonetheless, *PIK3CA* gene amplification showed a more prominent role, since it was detected in over 50% of laryngeal dysplasias, and 70% of the lesions that progressed to invasive carcinoma. More importantly, this study uncovers *PIK3CA* gene amplification as a significant independent predictor of laryngeal cancer development beyond the current histopathological criteria.

Multiple genomic alterations in the *PIK3CA* gene (e.g., mutations, amplifications, and overexpression) have been widely documented in HNSCC and impact around 55% of cases [42]. In line with this, Shu-Chun et al. reported that a higher *PIK3CA* copy number was associated with the increased likelihood of lymph node metastasis in oral carcinomas. In addition, they found a gradual increase in the copy number of *PIK3CA* and other genes (i.e., *TERC* and *ZASC1*) from non-lesional states to more advanced lesions [14]. *PIK3CA* amplification has also been found to be associated with poor prognosis in HNSCC patients without lymph node metastasis, and therefore could be a potential prognostic marker [43]. Moreover, the *PIK3CA* gene is among the most frequently mutated in the HNSCC oncogenome [10], and the hot-spot mutations E542K, E545K, and H1047R have shown high oncogenic potential and therapy-resistant phenotypes [44].

PI3K signaling is active in over 90% of HNSCCs, which may also provide excellent therapeutic opportunities. PI3K pathway activation has been related to resistance to radiotherapy and chemotherapeutic drugs such as cisplatin, 5-FU, and paclitaxel [45]. In addition, pharmacologic inhibitors of key pathway components have shown remarkable effects on tumor cell growth and radiotherapy sensitization in preclinical models, which has prompted the design of several combination clinical trials in HNSCCs and other solid tumors [45]. Furthermore, the PIK3CA-specific inhibitor alpelisib holds promising potential to effectively target and ameliorate uncontrolled cell growth and survival in tumors harboring *PIK3CA* mutation and/or gene amplification [46]. In light of these data, we can speculate on the possibility of using alpelisib as a molecular-targeted therapy for HNSCC patients and as a prophylactic treatment for HNSCC prevention. Our study significantly contributes to improve knowledge on the early prevalent genetic alterations present in laryngeal precancerous lesions. These results may also pave the way toward developing new methods of risk stratification and to guide early treatment interventions.

Regarding *SOX2* gene amplification, this was observed in a smaller proportion of laryngeal dysplasias (31%); all positive cases except one harbored co-amplification of the *PIK3CA* gene. Interestingly, the combination of *PIK3CA* and *SOX2* amplification allowed us to distinguish three different cancer risk subgroups: patients who carried amplification of both genes exhibited the highest risk of progression to laryngeal carcinoma, whereas patients with amplification of a single gene showed an intermediate risk, and the amplification-negative subgroup had the lowest cancer risk. These findings indicate that the combined amplification of *PIK3CA* and *SOX2* could enhance the early cancer detection and risk stratification of patients with laryngeal precancerous lesions. Therefore, this simple PCR-based test could be easily implemented in clinical practice in order to increase the precision of current histopathological diagnosis.

A cooperative function betweenSOX2 and the PI3K signaling pathway has been demonstrated in squamous lung cancer [47]. Given that smoking is a primary etiological factor commonly shared in aerodigestive tract cancers, it is plausible that this mechanism also operates in laryngeal cancer pathogenesis to promote malignant progression through the 3q26 amplicon. It is worth mentioning that the amplification of genes at 3q26 can also be detected in brush samples [47], plasma ctDNA [48], and liquid-based cytology specimens [49]. Therefore, 3q26 gene amplification could be used to individualize patient risk stratification and develop precision HNSCC screening and prevention strategies, which currently do not exist for this disease.

Current cancer detection innovations intend to integrate multimodal information from clinical, histological, radiological, molecular, and/or environmental data to generate cancer risk stratification profiles that allow for the accurate prediction of high-risk individuals and personalized treatment decisions. Efforts for early cancer detection should also focus on avoiding adverse outcomes such as overdiagnosis and overtreatment.

Our study holds significant value due to the inherent challenge in detecting premalignant lesions in the larynx, which makes each case analyzed extremely valuable and informative. Nevertheless, the retrospective nature of this study and a relatively small sample size should be acknowledged as its main limitations. These findings should be further confirmed in large prospective validation studies.

## 4. Materials and Methods

### 4.1. Patients and Tissue Specimens

Surgical tissue specimens from patients who were diagnosed with laryngeal dysplasia at the Hospital Universitario Central de Asturias between 1996 and 2010 were retrospectively collected according to approved institutional review board guidelines. This study was performed in accordance with the principles of the Declaration of Helsinki with the appropriate approval of the Ethical and Scientific Committees of the Hospital Universitario Central de Asturias and the Regional CEIm from Principado de Asturias (date of approval 14 May 2019; approval number: 141/19, for the project PI19/00560). Informed consent was obtained from all patients.

A total of 62 patients were selected for study. All patients were men, with a mean age of 65 years (ranging from 36 to 83 years). A total of 8 (13%) premalignant lesions were classified as low-grade dysplasia and 54 (87%) as high-grade dysplasia according to the WHO classification (4th Edition) [29,30].

The excisional biopsy of lesions was carried out either with a CO_2_ laser or with cold instruments. A complete macroscopic exeresis of the lesion was undertaken in all cases, but the microscopic margins were not addressed. Patients were followed up every two months in the first six months after completing the treatment, every three months until the second year, and every six months thereafter.

Archival, paraffin-embedded blocks from the original tissue biopsies were provided by the Principado de Asturias BioBank (PT20/00161), which is part of the Spanish National Biobanks Network.

### 4.2. Inclusion and Exclusion Criteria

Patients needed to meet the following inclusion criteria to be enrolled in this study: (1) Pathologic diagnosis of laryngeal dysplasia. (2) Lesions of the vocal folds. (3) No previous history of head and neck cancer. (4) Complete excisional biopsy of the lesion. (5) A minimum follow-up of five years (or until progression to malignancy). Patients with a diagnosis of laryngeal dysplasia who developed cancer within the next six months were excluded from the study.

### 4.3. Gene Amplification Analysis by Real-Time PCR (qPCR)

The protocol for DNA extraction from paraffin-embedded tissue sections has been previously described [50]. Briefly, formalin-fixed paraffin-embedded (FFPE) tissue samples were subjected to thorough deparaffinization with xylene (Sigma Aldrich, Saint Louis, MO, USA) and methanol (Merck, Darmstadt, Germany) washings to remove all traces of the xylene, and 24 h incubation in 1 mol/L sodium thiocyanate (Sigma Aldrich Inc.) to reduce cross-links. Subsequently, the tissue pellet was digested for 2–3 days in lysis buffer with high doses of proteinase K (final concentration, 2 µg/µL, freshly added twice daily). Finally, DNA extraction was performed using the QIAamp DNA Mini Kit (Qiagen GmbH, Hilden, Germany). DNA extracted from normal mucosa obtained from non-oncologic patients was used as a calibrator sample. Gene amplification was evaluated by qPCR in a StepOne Plus (Applied Biosystems, Foster City, CA, USA) using Power SYBR Green PCR Master Mix and oligonucleotides with the following sequences: for the *PIK3CA* gene, Fw, 5′-TTTTGCATTTTTATCTATCAGTCCA-3′ and Rv, 5′-TGCATTTTAATGGTGGAGAGG-3′; for the *SOX2* gene, Fw, 5′-CTCCGGGACATGATCAGC-3′ and Rv, 5′-CTGGGACATGTGAAGTCTGC-3′; and for the reference gene, *COL7A1* (located at 3p21), Fw, 5′-ACCCAGTACCGCATCATTGTG-3′ and Rv, 5′-TCAGGCTGGAACTTCAGTGTGT-3′. Reactions were carried out with 6.25 ng DNA template using primer concentrations of 300 nM (Fw) and 300 nM (Rv) or 900 nM (Rv) for *COL7A1*. Samples were analyzed in triplicate and template-free blanks were also included. The relative gene copy number for *PIK3CA* and *SOX2* was calculated using the 2^−ΔΔCT^ method. Calibration curves for the reference gene (*COL7A1*) and the target genes (either *PIK3CA* or *SOX2*) showed parallel slopes and comparable amplification efficiency across the linear range (25–1.5 ng) (Appendix A). The ΔΔC_T_ represents the difference between the ΔC_T_ of dysplasia and ΔC_T_ of normal mucosa, with ΔC_T_ being the average C_T_ for the target gene (*PIK3CA* or *SOX2*) minus the average C_T_ for the reference gene (*COL7A1*). The optimal cut-off value for gene amplification was set at 1.75 calculated by ROC curve analysis using as the end point the progression to cancer. Values > 1.75 were considered positive amplification.

### 4.4. Immunohistochemistry

Paraffin-embedded tissue samples were cut into 3-µm sections, deparaffinized with standard xylene, and hydrated through graded alcohols into water. Antigen retrieval was performed by heating the sections with Envision Flex Target Retrieval solution, high pH (Dako, Glostrup, Denmark). Staining was carried out at room temperature on an automatic staining workstation (Dako Autostainer Plus, Glostrup, Denmark) with the following primary antibodies: PI3 Kinase p110α rabbit monoclonal antibody at 1:100 dilution (C73F8; Cell Signaling Technology #4249) or anti-SOX2 rabbit polyclonal antibody at 1:1000 dilution (Merck Millipore, #AB5603) using the Dako EnVision Flex + Visualization System (Dako Autostainer, Glostrup, Denmark) and diaminobenzidine chromogen as the substrate. The final step was counterstaining with hematoxylin.

### 4.5. Statistical Analysis

The χ^2^ and Fisher’s exact tests were used for comparison between the categorical variables. The correlation between the amplification of the *PIK3CA* and *SOX2* genes was assessed using the Spearman coefficient. For time-to-event analysis, Kaplan–Meier curves were plotted. Differences between survival times were analyzed by the log-rank method. Univariable and multivariable Cox regression analyses were performed. The hazard ratios (HR) with 95% confidence interval (CI) and *p* values were reported. The predictive potential of the studied variables was evaluated by performing receiver operating characteristic curve (ROC) analysis, and the discriminative efficacy of the individual variable was calculated by the area under the ROC curve (AUC). All tests were two-sided. *p* values of ≤0.05 were considered statistically significant.

## 5. Conclusions

This study demonstrates the clinical relevance of *PIK3CA* and *SOX2* amplification in laryngeal tumorigenesis as early cancer risk markers beyond current histopathological grading. *PIK3CA* gene amplification exhibits a more prominent role; it was detected in over 70% of progressing dysplasias, and unprecedentedly uncovered as a significant independent predictor of laryngeal cancer development. Furthermore, combined amplification of *PIK3CA* and *SOX2* is emerging as a valuable and easy-to-implement tool for cancer risk stratification that may complement histopathological diagnosis to distinguish high-risk patients more accurately, and to ultimately improve personalized treatment decisions.

## Figures and Tables

**Figure 1 ijms-25-02695-f001:**
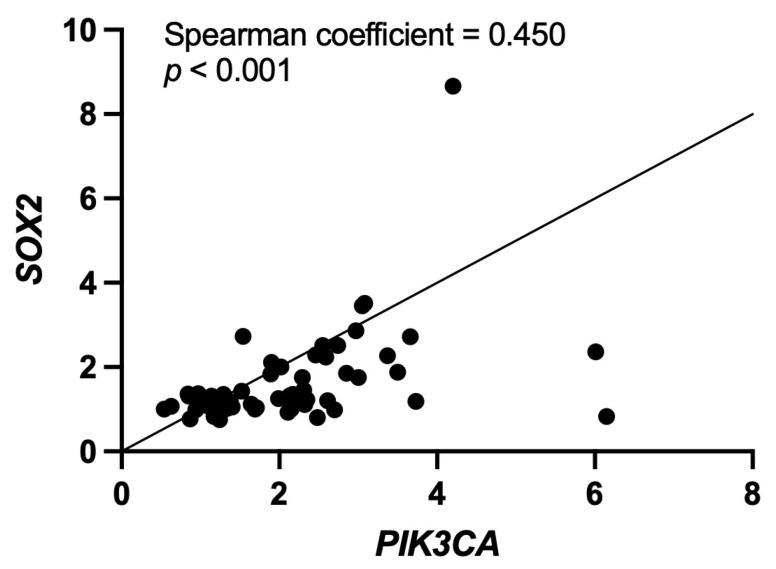
*PIK3CA* and *SOX2* gene amplification analysis by real-time PCR in patients with laryngeal precancerous lesions. Spearman correlation between the relative gene copy numbers calculated for *PIK3CA* and *SOX2*.

**Figure 2 ijms-25-02695-f002:**
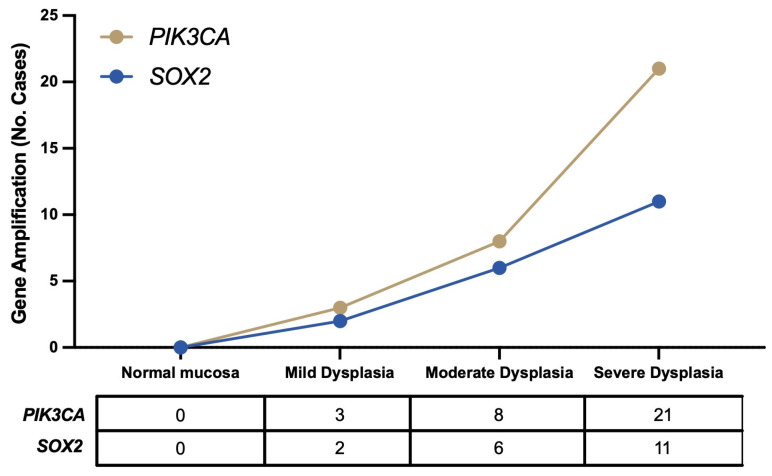
Frequency comparison of *PIK3CA* and *SOX2* gene amplification along the different stages of laryngeal tumorigenesis. The graph represents the percentage of positive cases harboring amplification of *PIK3CA* or *SOX2* genes, as detected by qPCR.

**Figure 3 ijms-25-02695-f003:**
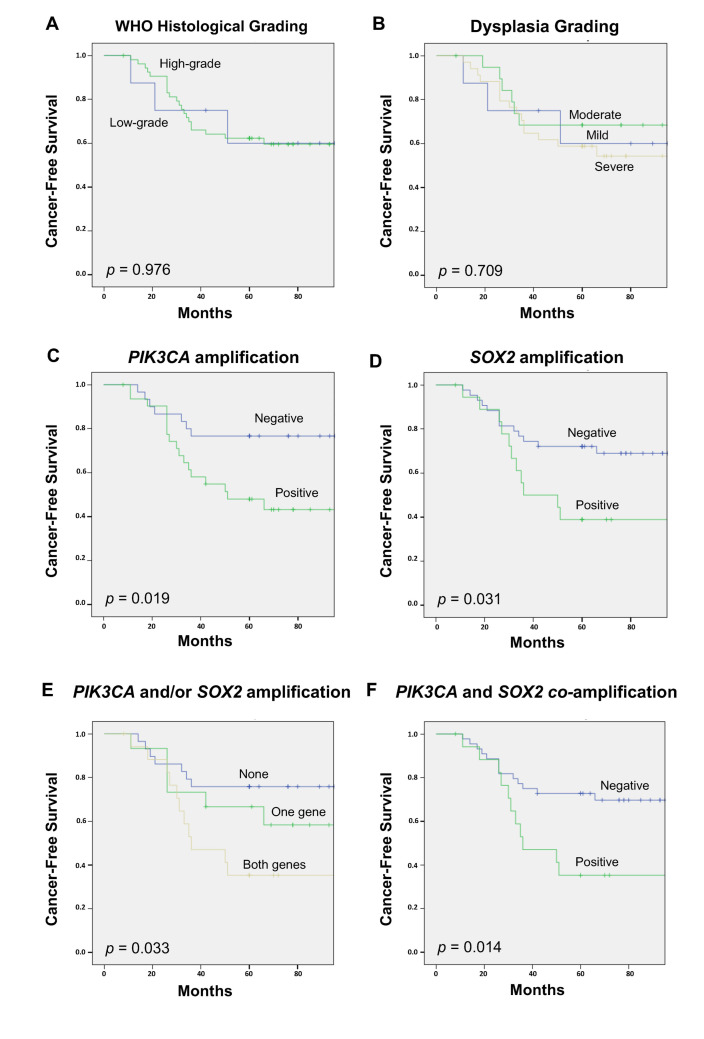
Kaplan–Meier cancer-free survival curves in the study series of 62 patients with laryngeal dysplasias categorized by WHO histological grading (**A**), dysplasia grading (**B**), *PIK3CA* gene amplification (**C**), *SOX2* gene amplification (**D**), *PIK3CA* and/or *SOX2* gene amplification (**E**), and *PIK3CA* and *SOX2* co-amplification (**F**). *p* values were estimated using the log-rank test.

**Table 1 ijms-25-02695-t001:** Crosstab to evaluate the correlation between *PIK3CA* and *SOX2* gene amplification in 62 laryngeal dysplasias.

Characteristic	Negative *PIK3CA* Amplification	Positive *PIK3CA* Amplification	Total Cases
Negative *SOX2* Amplification	29	14	43
Positive *SOX2* Amplification	1	18	19
Total Cases	30	32	62

**Table 2 ijms-25-02695-t002:** Evolution of premalignant lesions in relation to histopathological diagnosis, *PIK3CA*, and *SOX2* gene amplification.

Characteristic	No. of Cases (%)	Progression to Carcinoma (%)	*p*
**Age** <55 years 55–70 years >70 years	14 (22) 24 (39) 24 (39)	6 (43) 11 (46) 7 (29)	0.464 *
**Tobacco Smoking** No Yes	50 (82) 11 (18)	21 (42) 3 (27)	0.502 ^†^
**WHO Histopathological diagnosis** Low-grade dysplasia High-grade dysplasia	8 (13) 54 (87)	3 (37) 21 (39)	1.000 ^†^
**Dysplasia grade** Mild dysplasia Moderate dysplasia Severe dysplasia	8 (13) 20 (32) 34 (55)	3 (37) 6 (30) 15 (44)	0.588 *
***PIK3CA* gene amplification** Negative Positive	30 (48) 32 (52)	7 (23) 17 (53)	0.020 ^†^
***SOX2* gene amplification** Negative Positive	43 (69) 19 (31)	13 (30) 11 (58)	0.051 ^†^
***PIK3CA* and/or *SOX2* amplification** Negative One positive Both positive	29 (47) 15 (24) 18 (29)	7 (24) 6 (40) 11 (61)	0.040 *

^†^ Fisher’s exact test and * Chi-square test.

**Table 3 ijms-25-02695-t003:** Receiver operating characteristic (ROC) analysis for individual and combined amplification of the *PIK3CA* and *SOX2* genes.

Characteristic	AUC	95% CI	*p*
*PIK3CA* gene amplification	0.657	0.517–0.797	0.039
*SOX2* gene amplification	0.624	0.477–0.771	0.102
*PIK3CA* and *SOX2* amplification	0.675	0.535–0.815	0.021

AUC: Area under the curve; 95% CI: 95% confidence interval (CI).

## Data Availability

Data available upon request to the corresponding author (JMG-P) due to privacy/ethical restrictions.

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
