# Peer review of "Combined PIK3CA and SOX2 Gene Amplification Predicts Laryngeal Cancer Risk beyond Histopathological Grading"

_ijms, 2024, doi:10.3390/ijms25052695_

Round 1

Reviewer 1 Report

Comments and Suggestions for Authors

Montoro-Jimenez and colleagues proposed a research article aimed at evaluating the clinical significance of PIK3CA and SOX2 gene amplification in early laryngeal tumorigenesis, by detecting these mutations in precancerous lesions. For this purpose, the authors analyzed 62 laryngeal precancerous lesions founding both SOX2 (31%) and PIK3CA (52%) gene amplifications, with co-amplification in 29% of cases. Then the authors analyzed patients’ clinical features revealing that PIK3CA amplification is associated with a worse prognosis. Overall, the research idea is interesting, however, the experimental design is too simple and no functional experiments were performed. Below are reported some minor/major comments that will improve the quality of the study:

1) In the Introduction section, to better emphasize the relevance of head and neck cancer, please briefly introduce the global burden of this pathology taking into account the latest data obtained by international consortia like the Global Burden Disease. For this purpose, please see:

- PMID: 37676656

- PMID: 37367741

- PMID: 36369568

- PMID: 35285945

2) The results obtained through qPCR were not confirmed by using other methods, like FISH or immunohistochemistry investigation. This represents a key limitation of the study. The authors have to perform such confirmatory analyses at least in a representative case series of positive samples;

3) You should enrich Table 1 by performing other statistical analyses stratifying patients according to age groups (e.g. <45 years old; 45 < years old < 65; > 65 years old);

4) In the Introduction or Discussion section, please briefly introduce both SOX2 and PIK3CA describing why these genes are involved in cancer development and progression when altered. In particular, please mention their involvement in cancer progression and drug resistance (demonstrated in multiple tumors) as well as the importance of these mutations for the development of tailored approaches (e.g. by using PIK3CA inhibitors like alpelisib). For this purpose, please see:

- PMID: 35335966

- PMID: 30380422

- PMID: 35357905

- PMID: 31600013

Author Response

Comments and Suggestions for Authors

Montoro-Jimenez and colleagues proposed a research article aimed at evaluating the clinical significance of PIK3CA and SOX2 gene amplification in early laryngeal tumorigenesis, by detecting these mutations in precancerous lesions. For this purpose, the authors analyzed 62 laryngeal precancerous lesions founding both SOX2 (31%) and PIK3CA (52%) gene amplifications, with co-amplification in 29% of cases. Then the authors analyzed patients’ clinical features revealing that PIK3CA amplification is associated with a worse prognosis. Overall, the research idea is interesting, however, the experimental design is too simple and no functional experiments were performed.

Below are reported some minor/major comments that will improve the quality of the study:

Response: We thank the reviewer for considering that our study is interesting and for his/her insightful recommendations.

1) In the Introduction section, to better emphasize the relevance of head and neck cancer, please briefly introduce the global burden of this pathology taking into account the latest data obtained by international consortia like the Global Burden Disease. For this purpose, please see:

- PMID: 37676656

- PMID: 37367741

- PMID: 36369568

- PMID: 35285945

Response: We agree. Information regarding the global burden of this disease has now been included in the Introduction alongside the suggested references. The only exception was PMID: 35285945, which was not included because it this reference is a Comment but not an article.

2) The results obtained through qPCR were not confirmed by using other methods, like FISH or immunohistochemistry investigation. This represents a key limitation of the study. The authors have to perform such confirmatory analyses at least in a representative case series of positive samples;

Response: It is worth to note that gene copy number assessment by qPCR is well accepted as a sensitive, reliable and easy-to-implement method. As such, this methodology has been extensively and widely used in our laboratory (and others) to evaluate gene amplification in different tissue samples from HNSCC patients, precancerous lesions as well as HNSCC-derived cell lines. Please find below some examples from our published studies, where gene copy numbers obtained by qPCR were found to correlate with mRNA and/or protein expression levels. Nevertheless, since the reviewer considers that the lack of confirmation for our qPCR results by another method is a key study limitation, we have performed immunohistochemical analysis to confirm detection of both proteins in various cases harboring amplification of PIK3CA and SOX2 genes. This new information is shown in Supplementary Figure S2, and also described in the text of Results and Methods.

References

  1. Rodrigo JP, García-Carracedo D, García LA, Menéndez S, Allonca E, González MV, Fresno MF, Suárez C, García-Pedrero JM. Distinctive clinicopathological associations of amplification of the cortactin gene at 11q13 in head and neck squamous cell carcinomas. J Pathol. 2009 Mar;217(4):516-23. doi: 10.1002/path.2462. PMID: 18991334.
  2. Rodrigo JP, Álvarez-Alija G, Menéndez ST, Mancebo G, Allonca E, García-Carracedo D, Fresno MF, Suárez C, García-Pedrero JM. Cortactin and focal adhesion kinase as predictors of cancer risk in patients with laryngeal premalignancy. Cancer Prev Res (Phila). 2011 Aug;4(8):1333-41. doi: 10.1158/1940-6207.CAPR-10-0338. Epub 2011 Jun 6. PMID: 21646305.
  3. Hermida-Prado F, Menéndez ST, Albornoz-Afanasiev P, Granda-Diaz R, Álvarez-Teijeiro S, Villaronga MÁ, Allonca E, Alonso-Durán L, León X, Alemany L, Mena M, Del-Rio-Ibisate N, Astudillo A, Rodríguez R, Rodrigo JP, García-Pedrero JM. Distinctive Expression and Amplification of Genes at 11q13 in Relation to HPV Status with Impact on Survival in Head and Neck Cancer Patients. J Clin Med. 2018 Dec 1;7(12):501. doi: 10.3390/jcm7120501. PMID: 30513772; PMCID: PMC6306931.
  4. Menéndez ST, Villaronga MA, Rodrigo JP, Alvarez-Teijeiro S, García-Carracedo D, Urdinguio RG, Fraga MF, Pardo LA, Viloria CG, Suárez C, García-Pedrero JM. Frequent aberrant expression of the human ether à go-go (hEAG1) potassium channel in head and neck cancer: pathobiological mechanisms and clinical implications. J Mol Med (Berl). 2012 Oct;90(10):1173-84. doi: 10.1007/s00109-012-0893-0. Epub 2012 Mar 31. PMID: 22466864.

3) You should enrich Table 1 by performing other statistical analyses stratifying patients according to age groups (e.g. <45 years old; 45 < years old < 65; > 65 years old);

Response: Following the reviewer’s recommendation, we have performed the suggested analyses stratifying patients by age. These new data have been added to Table 1 (now renumbered to Table 2) and also the corresponding Kaplan-Meier cancer-free survival curves as new Supplementary Figure S3. Since there were no patients < 45 years in our study cohort, patients were distributed according to the age terciles (i.e. <55 years; 55-70 years; >70 years). These data clearly show no association between age and the risk of progression to laryngeal carcinoma, as now stated in the text of Results.

4) In the Introduction or Discussion section, please briefly introduce both SOX2 and PIK3CA describing why these genes are involved in cancer development and progression when altered. In particular, please mention their involvement in cancer progression and drug resistance (demonstrated in multiple tumors) as well as the importance of these mutations for the development of tailored approaches (e.g. by using PIK3CA inhibitors like alpelisib). For this purpose, please see:

- PMID: 35335966

- PMID: 30380422

- PMID: 35357905

- PMID: 31600013

Response: The Introduction has been further enriched by providing extensive evidences on the implications of PIK3CA and SOX2 alterations in tumor development, disease progression and drug resistance, according to the reviewer’s suggestion. On the other hand, the importance of PIK3CA and SOX2 amplification for the development of tailored strategies for cancer prevention and treatment has also been discussed in our new version of the manuscript.

Reviewer 2 Report

Comments and Suggestions for Authors

Dear Authors,

It was a pleasure to read your article. I believe your paper might be interesting to readers from the clinical field. Your paper is well written and organized.

However, there are some scopes to improve the quality of the manuscript. The reviewer would like to suggest the following revision in the manuscript. 

The aim of this technical note "
Combined PIK3CA and SOX2 gene amplification predicts laryngeal cancer risk beyond histopathological grading" was to evaluate the clinical significance of PIK3CA and SOX2 gene amplification in early stages of HNSCC tumorigenesis, thereby evaluating their predictive value for cancer risk assessment in patients with laryngeal precancerous lesions. 

English language fine. No issues detected.
Punctuation should be corrected.

Standardize text structure and alignment according to guidelines.

Authors list: Please remove academic titles.

Abstract:
Key words: SOX2, PIK3CA should be written in italics.

Introduction 
lines 53-59: Expand information on the oncogenicity of this pathway. What effects does it cause? What disturbances in the activity of this pathway influence cancer progression? Add this references:

Mishra, R.; et al. PI3K Inhibitors in Cancer: Clinical Implications and Adverse Effects. Int. J. Mol. Sci. 202122, 3464. https://doi.org/10.3390/ijms22073464

Starzyńska, A.; et al. Any Role of PIK3CA and PTEN Biomarkers in the Prognosis in Oral Squamous Cell Carcinoma? Life 202010, 325. https://doi.org/10.3390/life10120325

Materials and Methods 

Add inclusion and exclusion criteria as separate subsections.

line 89: Don't start a sentence with a number! Improve throughout the text.

line 96: The authors are encouraged to discuss in detail the gene amplification analysis by Real-time PCR (qPCR) procedure.

Results 

Prepare tables according to the guidelines.

Figure 1B add as separate table.

The quality of the figures is poor. Increase quality.

Discussion

lines 209-213: add reference: Gale N, Poljak M, Zidar N. Update from the 4th Edition of the World Health Organization Classification of Head and Neck Tumours: What is New in the 2017 WHO Blue Book for Tumours of the Hypopharynx, Larynx, Trachea and Parapharyngeal Space. Head Neck Pathol. 2017 Mar;11(1):23-32. doi: 10.1007/s12105-017-0788-z. 

In your discussion, discuss the limitations of the study, including the small sample size. Add the advantages of the study. Discuss future opportunities for researchers and research directions. 

line 315: Authors’ Contributions : Use abbreviations of your surname and first name.

Add a table with abbreviations before references.

Reconsider after major revision

Comments on the Quality of English Language

Minor editing of English language required

Author Response

Comments and Suggestions for Authors

Dear Authors,

It was a pleasure to read your article. I believe your paper might be interesting to readers from the clinical field. Your paper is well written and organized.

Response: We thank the reviewer for highlighting the interest of our study and also manuscript quality.

However, there are some scopes to improve the quality of the manuscript. The reviewer would like to suggest the following revision in the manuscript. The aim of this technical note "Combined PIK3CA and SOX2 gene amplification predicts laryngeal cancer risk beyond histopathological grading" was to evaluate the clinical significance of PIK3CA and SOX2 gene amplification in early stages of HNSCC tumorigenesis, thereby evaluating their predictive value for cancer risk assessment in patients with laryngeal precancerous lesions.

Response: We thank the reviewer for his/her meticulous revision and valuable suggestions.

English language fine. No issues detected.

Punctuation should be corrected.

Response: This has now been corrected.

Standardize text structure and alignment according to guidelines.

Response: Text structure has been adjusted according to the guidelines.

Authors list: Please remove academic titles.

Response: Academic titles have now been removed.

Abstract:

Key words: SOX2, PIK3CA should be written in italics.

Response: These key words have been corrected and written in italics.

Introduction

lines 53-59: Expand information on the oncogenicity of this pathway. What effects does it cause? What disturbances in the activity of this pathway influence cancer progression? Add this references:

Mishra, R.; et al. PI3K Inhibitors in Cancer: Clinical Implications and Adverse Effects. Int. J. Mol. Sci. 202122, 3464. https://doi.org/10.3390/ijms22073464

Starzyńska, A.; et al. Any Role of PIK3CA and PTEN Biomarkers in the Prognosis in Oral Squamous Cell Carcinoma? Life 202010, 325. https://doi.org/10.3390/life10120325

Response: The Introduction has been extended with additional data and references supporting the oncogenic role of PIK3CA and SOX2 and the impact of their altered expression/function. In addition, the two suggested references have been added.

Materials and Methods

Add inclusion and exclusion criteria as separate subsections.

Response: Inclusion and exclusion criteria have been moved to a separate subsection.

line 89: Don't start a sentence with a number! Improve throughout the text.

Response: The text has been checked and accordingly amended.

line 96: The authors are encouraged to discuss in detail the gene amplification analysis by Real-time PCR (qPCR) procedure.

Response: Full details on the protocols for DNA extraction and gene amplification analysis by qPCR have been included in the revised manuscript.

Results

Prepare tables according to the guidelines.

Response: Tables are prepared according to the guidelines.

Figure 1B add as separate table.

Response: Figure 1B has been removed and added as new Table 1 in this new version of the manuscript. Please note that subsequent Tables have been renumbered (now Tables 2 and 3).

The quality of the figures is poor. Increase quality.

Response: The quality of Figures 1-3 has been improved by replacing all previous images with others at higher resolution.

Discussion

lines 209-213: add reference: Gale N, Poljak M, Zidar N. Update from the 4th Edition of the World Health Organization Classification of Head and Neck Tumours: What is New in the 2017 WHO Blue Book for Tumours of the Hypopharynx, Larynx, Trachea and Parapharyngeal Space. Head Neck Pathol. 2017 Mar;11(1):23-32. doi: 10.1007/s12105-017-0788-z.

Response: This reference has been added, as suggested.

In your discussion, discuss the limitations of the study, including the small sample size. Add the advantages of the study. Discuss future opportunities for researchers and research directions.

Response: The main study limitations and advantages have been discussed.

line 315: Authors’ Contributions : Use abbreviations of your surname and first name.

Response: These corrections have been made.

Add a table with abbreviations before references.

Response: A table with the abbreviations has been added before the references.

Reconsider after major revision

Reviewer 3 Report

Comments and Suggestions for Authors

The manuscript highlights the genomic of  PIK3CA and SOX in Laryngeal cancer management.

However the data is not robust and the discussion segment is weak.

Other comments are in yellow highlight in the attached text

Comments on the Quality of English Language

English is acceptable with minor revision needed.

Author Response

Comments and Suggestions for Authors

The manuscript highlights the genomic of PIK3CA and SOX in Laryngeal cancer management.

Response: Quite remarkably, our study unprecedentedly demonstrates that PIK3CA and SOX2 amplification assessment may serve as a valuable and easy-to-implement tool for cancer risk stratification to complement/improve the still limited predictability of current WHO histopathological diagnosis, gold standard in the clinical routine.

However the data is not robust and the discussion segment is weak.

Response: We honestly think that our study provides highly relevant findings and of major interest and applicability potential for early cancer risk evaluation in patients with laryngeal precancerous lesions. Moreover, since 3q26 amplification is common in other cancers, our results could also provoke analogous studies to investigate and prove PIK3CA and SOX2 amplification as early cancer risk biomarkers to complement histological diagnosis and stratification of patients with other precancerous lesions (e.g. oral leukoplakia, CIN…).

On the other hand, the extensive new data and information added in this revised version of the manuscript should hopefully contribute to further strengthen our manuscript and findings. In this regard, we are deeply grateful to the reviewers for their insightful recommendations, which have undoubtedly helped to improve our work.

Other comments are in yellow highlight in the attached text

Response: Please note that all the additional comments in relation to the text highlighted in the pdf file attached by the reviewer have been listed below, accompanied by our responses to each comment or question raised.

Introduction

“…tumor condition and behavior…” This is unclear, please revise.

Response: This has been amended in our new version of the manuscript.

Methods

8 (13%) premalignant lesions were classified as low-grade dysplasia and 54 (87%) as following the WHO classification (4th Edition) [15]”. This statement is incorrect, please revise.

Response: This sentence has now been amended.

The protocol for DNA extraction from paraffin-embedded tissue sections has been 96 described elsewhere [16]”. Where/what is ‘elsewhere’? Please revise.

Response: To clarify this, the term ‘elsewhere’ has been removed and this text is now changed. In addition, further details on the DNA extraction procedure have been added.

Results

PIK3CA and SOX2 gene copy amplification in early stages of laryngeal tumorigenesis Amplification of SOX2 and PIK3CA genes was frequently detected in 19 (31%) and 32 (52%) out of 62 laryngeal dysplasias, respectively. Co-amplification of both genes was present in 18 (29%) cases. The relative copy numbers ranged from 2- to 6-fold for PIK3CA amplification, and from 2- to 9-fold for SOX2 amplification. In addition, a strong positive correlation was observed between the amplification of PIK3CA and SOX2 genes (Figure 1; p < 0.001; Spearman coefficient = 0.450). Interestingly, gene amplification was found to gradually increase along the early stages of laryngeal tumorigenesis, as shown in Figure 2; however, PIK3CA amplification occurred at a higher frequency reaching over 50% of cases with severe dysplasia”. The data is small and insignificant.

Response: We cannot agree with this statement. PIK3CA and SOX2 amplifications were indeed frequently detected in 31-52% laryngeal dysplasias. More importantly, our study demonstrates the clinical relevance of amplification of both genes as robust predictors of laryngeal cancer risk, beyond current WHO histopathological classification. Therefore, altogether these data clearly show that are not insignificant at all.

PIK3CA and SOX2 amplifications increased with the grade of dysplasia, and both were predominantly detected in high-grade dysplasias. Thus, SOX2 gene amplification was observed in 2/8 (25%) low-grade dysplasias and 17/54 (31%) high-grade dysplasias. In turn, PIK3CA gene amplification was detected at a higher frequency in 3/8 (37.5%) low- grade dysplasias and 29/54 (54%) high-grade dysplasias. However, the differences did not reach statistical significance (p = 1.00 and p = 0.467, respectively; Fisher’s exact test)”. This is again a small and insignificant data.

Response: We again disagree on this statement. We found very strong significant correlations between PIK3CA and SOX2 amplifications and the risk of progression, as shown in Table 2 (previous Table 1) and also the Kaplan Meier cancer-free survival graphs in Figure 3, and by ROC in Table 3. Furthermore, PIK3CA gene amplification was also found a significant independent predictor in multivariable Cox analyses (HR = 2.74, 95%CI 1.13-6.66, p = 0.025), as indicated in Results. The lack of significant correlation between PIK3CA and SOX2 amplification and the grade of dysplasia could be related to the limited predictive value of the histological grading in our study cohort. Notably, the current WHO 2-tier classification (i.e. high-grade and low-grade dysplasia) and the previous 3-tier dysplasia grading (i.e. mild, moderate and severe dysplasia) both failed to significantly predict laryngeal cancer risk, as now shown in Table 2 and Figure 3A and B.

Please also note that data on Table 2 have now been extended by adding patient stratification by age, as recommended by Reviewer #1 (see Point 3). In addition, the corresponding Kaplan-Meier cancer-free survival curves have also been added as new Supplementary Figure S3. Our data show no association between age and the risk of progression to laryngeal carcinoma, as now stated in the text of Results.

Discussion

“…early detection biomarkers”. This is unclear.

Response: This has been amended for clarification.

 “A new WHO classification has recently been established [15]”. Please insert this WHO classifications details.

Response: Details for this new WHO classification have been added, and also a new reference suggested by Reviewer #2 (comment on lines 209-213).

That attempts to overcome/improve the limited predictability of the previous 3-tier dysplasia grading system”. This is incorrect statement, please revise.

Response: We apologize for this error, which has been corrected in our revised version of the manuscript.

In line with this, Shu-Chun et al. reported that a higher PIK3CA copy number was associated with increased likelihood of lymph node metastasis in oral carcinomas. In addition, they also found a gradual increase in the copy number of PIK3CA and other genes (i.e. TERC and ZASC1) from a non-lesional state to more advanced lesions [6]”. Any literatures on laryngeal cancer?

Response: We and others have investigated the role of PIK3CA and other PI3K pathway alterations in different HNSCC sites (including laryngeal cancer) (please see references below). However, to the best of our knowledge, this is the first study that addresses the role of PIK3CA amplification in laryngeal precancerous lesions, unprecedentedly uncovering a relationship with the risk of progression to invasive carcinoma.

Regarding the reported data in the literature, PIK3CA amplification has been found to associate with poor prognosis in HNSCC patients without lymph node metastasis, and hence pointed as a prognostic marker (PMID: 22994622). Moreover, PIK3CA gene is among the most frequently mutated in HNSCC oncogenomes (PMID: 25631445), and the hot-spot mutations E542K, E545K and H1047R have demonstrated high oncogenic potential. Besides, PI3K signaling is active in over 90% of HNSCC, which may also offer excel-lent therapeutic opportunities. PI3K pathway activation has been related to resistance to radiotherapy and also chemotherapeutic drugs, such as cisplatin, 5-FU and paclitaxel (PMID: 31785230). In addition, pharmacologic inhibitors of key pathway components have shown remarkable effects on tumor cell growth and radiotherapy sensitization in preclinical models, which prompted to design several combination clinical trials in HNSCC and other solid tumors (PMID: 31785230). Furthermore, the PIK3CA-specific inhibitor alpelisib holds promising potential to effectively target and ameliorate uncontrolled cell growth and survival in tumors harboring PIK3CA mutation and/or gene amplification (PMID: 35357905). In the light of these data, we could speculate with the possibility of using alpelisib as a molecular-targeted therapy for HNSCC patients and also a prophylactic treatment for HNSCC prevention. Accordingly, our findings have significantly contributed not only to improve knowledge on the early prevalent genetic alterations present in laryngeal precancerous lesions, but also to pave the way towards developing new methods of risk stratification and to guide early treatment interventions.

All the above-mentioned information has been included in the Discussion in the revised manuscript.

References:

  1. Cancer Genome Atlas Network. Comprehensive genomic characterization of head and neck squamous cell carcinomas. Nature. 2015 Jan 29;517(7536):576-82. doi: 10.1038/nature14129. PMID: 25631445.
  2. Suda T, Hama T, Kondo S, Yuza Y, Yoshikawa M, Urashima M, Kato T, Moriyama H. Copy number amplification of the PIK3CA gene is associated with poor prognosis in non-lymph node metastatic head and neck squamous cell carcinoma. BMC Cancer. 2012 Sep 20;12:416. doi: 10.1186/1471-2407-12-416. PMID: 22994622.
  3. García-Carracedo D, Villaronga MÁ, Álvarez-Teijeiro S, Hermida-Prado F, Santamaría I, Allonca E, Suárez-Fernández L, Gonzalez MV, Balbín M, Astudillo A, Martínez-Camblor P, Su GH, Rodrigo JP, García-Pedrero JM. Impact of PI3K/AKT/mTOR pathway activation on the prognosis of patients with head and neck squamous cell carcinomas. Oncotarget. 2016 May 17;7(20):29780-93. doi: 10.18632/oncotarget.8957. PMID: 27119232.
  4. Marquard FE, Jücker M. PI3K/AKT/mTOR signaling as a molecular target in head and neck cancer. Biochem Pharmacol. 2020 Feb;172:113729. doi: 10.1016/j.bcp.2019.113729. Epub 2019 Nov 27. PMID: 31785230.
  5. Du L, Chen X, Cao Y, Lu L, Zhang F, Bornstein S, Li Y, Owens P, Malkoski S, Said S, Jin F, Kulesz-Martin M, Gross N, Wang XJ, Lu SL. Overexpression of PIK3CA in murine head and neck epithelium drives tumor invasion and metastasis through PDK1 and enhanced TGFβ signaling. Oncogene. 2016 Sep 1;35(35):4641-52. doi: 10.1038/onc.2016.1. Epub 2016 Feb 15. PMID: 26876212.
  6. Li S, Huang XT, Wang MY, Chen DP, Li MY, Zhu YY, Yu Y, Zheng L, Qi B, Liu JQ. FSCN1 Promotes Radiation Resistance in Patients With PIK3CA Gene Alteration. Front Oncol. 2021 Jun 24;11:653005. doi: 10.3389/fonc.2021.653005. PMID: 34249689.

In the context of squamous lung cancer, it has been demonstrated a cooperative function of SOX2 and the PI3K signaling pathway, which is activated by tobacco smoking to trigger the squamous injury response in basal cells leading to hyperproliferation while preventing squamous differentiation [29]. Given that smoking is a primary etiological factor commonly shared in aerodigestive tract cancers, it is plausible that this mechanism operates in laryngeal cancer pathogenesis to promote malignant progression through 3q26 amplicon. It is also worth mentioning that amplification of genes at 3q26 can also be detected in brush samples from patients with oral leukoplakia and also oral mucosa samples from areca chewers without any apparent visible lesion [6]. There are also evidences demonstrating that 3q26 alterations can be useful and easily detectable in liquid biopsy and plasma ctDNA [30]. Of note, 3q26 gain has proved to robustly predict the progression of cervical dysplastic lesions using liquid-based cytology specimens [31]. Accordingly, 3q26 gene amplification points utility for individualized patient risk stratification and/or population-based cancer screening programs. Therefore, we could hypothesize that 3q26 amplicon could be used as the basis to develop precision HNSCC screening and prevention strategies, which currently do not exist for this disease. Nowadays, cancer detection innovations intend to integrate multimodal information from clinical, histological, radiological, molecular, and/or environmental data to generate cancer risk stratification profiles that allow accurate prediction of high-risk individuals and personalized treatment decisions. Efforts for early cancer detection should also focus on avoiding adverse outcomes, such as overdiagnosis and overtreatment”. The discussion should also include the humoural factors in the tumour microenvironment TME that is relevant to SOX2 & P13K and elaborates on the details of the genomic aspect of these

Response: As we described in the Introduction, HNSCCs (including laryngeal carcinomas) harbor many different chromosomal aberrations caused by carcinogen exposure. PIK3CA and SOX2 gene amplification at 3q26 region is among the most frequent recurrent genetic alterations in HNSCC as well as other cancers. By extension, our results demonstrate the frequency and early occurrence of these alterations in early stages of laryngeal cancer. Factors in the TME may influence the activation and function of SOX2 and PI3K signaling pathway; however, this should not play a role over gene amplification changes that occur at DNA level. Therefore, we consider that this aspect is unrelated to the study topic, and hence irrelevant to be discussed.

Nevertheless, Discussion and also Introduction sections have been extended by providing valuable additional information related to the oncogenic roles of PIK3CA and SOX2 genes in tumor progression, drug resistance and also potential therapeutic opportunities to tailor strategies for HNSCC treatment and/or prevention (e.g. using the PIK3CA inhibitor alpelisib), as detailed in our response to the previous comment.

Conclusions

This study demonstrates the clinical relevance of PIK3CA and SOX2 amplification in laryngeal tumorigenesis, as early cancer risk markers beyond current histopathological grading. PIK3CA gene amplification exhibits a more prominent role. Noteworthy, it was detected in over 70% of progressing dysplasias, and unprecedentedly uncovered as a significant independent predictor of laryngeal cancer development. Furthermore, combined amplification of PIK3CA and SOX2 emerges as a valuable and easy-to-implement tool for cancer risk stratification that may complement histopathological diagnosis to distinguish more accurately high-risk patients, and to ultimately improve personalized treatment decisions”. The conclusion is not strong.

Response: As we mentioned above, this study unprecedentedly demonstrates that PIK3CA and SOX2 amplification assessment may serve as a valuable and easy-to-implement tool for cancer risk stratification to complement/improve the still limited predictability of current WHO histopathological diagnosis. Therefore, we truly think that these are highly relevant findings and of major interest and applicability potential for early cancer risk evaluation in patients with laryngeal precancerous lesions.

References

The reference list is suboptimal.

Response: The reference list has been improved and extended considerably by including 16 additional references in this revised version of the manuscript.

Round 2

Reviewer 1 Report

Comments and Suggestions for Authors

Dear Authors,

thank you for your prompt reply. The revised version of the manuscript was significantly improved with new data and information correctly provided. 

I have only a further minor comment. In the Supplementary Figure S3 it seems that patients with > 70 y/o had a better cancer-free survival compared to younger individuals. Please give your comment to this result and provide a new survival analysis stratifying patients in <70 y/o and > 70 y/o. Please also check the labels in the figure with the content of the Figure legend were different age number are reported.

Author Response

Comments and Suggestions for Authors
Dear Authors,
thank you for your prompt reply. The revised version of the manuscript was
significantly improved with new data and information correctly provided.
Response: We thank the reviewer for considering that our manuscript has been
significantly improved by the new data included. We also reiterate our gratitude to this Reviewer for his/her insightful recommendations.

I have only a further minor comment. In the Supplementary Figure S3 it seems that patients with > 70 y/o had a better cancer-free survival compared to younger individuals. Please give your comment to this result and provide a new survival analysis stratifying patients in <70 y/o and > 70 y/o. Please also check the labels in the figure with the content of the Figure legend were different age number are reported.
Response: Following the reviewer’s recommendation, we have performed the
suggested new analysis dichotomizing patients’ age (<70 years vs >70 years). The corresponding Kaplan-Meier cancer-free survival curves have been included in Supplementary Figure S3, and commented in the text of Results. Figure legend has also been accordingly amended.

Reviewer 2 Report

Comments and Suggestions for Authors

The authors followed the reviewer's instructions. The article still requires minor editorial corrections. The article may be accepted after the Editor's decision.

Author Response

The authors followed the reviewer's instructions. The article still requires minor editorial corrections. The article may be accepted after the Editor's decision.
Response: We thank the reviewer for considering that all his/her recommendations were satisfactorily addressed and that our manuscript may be acceptable for publication. It is also worth noting that language editing of this new revision of our manuscript has been done by a native English speaker.

Reviewer 3 Report

Comments and Suggestions for Authors

The manuscript is not ready for publication. The discussion segment is weak.

The references list is suboptimal. Other comments are as in attached text.

Comments on the Quality of English Language

Minor editing of English is required.

Author Response

Comments and Suggestions for Authors
The manuscript is not ready for publication. The discussion segment is weak.
Response: We would like to remark that Discussion has considerably been extended by including meaningful information, which was requested by the reviewers upon the first revision round. Both Reviewer #1 and Reviewer #2 have no further caveats or concerns regarding our revised version of Discussion. Hopefully the additional changes in this new revised manuscript will be adequate and sufficient. Otherwise, we would appreciate if this reviewer could be more specific on explaining how we should strengthen this section and what information is missing. In relation to this, we should highlight that there are only few published studies assessing the contribution of 3q26 amplification in early tumorigenesis. Indeed, to the best of our knowledge, our study is the first to address the role of PIK3CA and SOX2 amplification in laryngeal cancer risk,
hence supporting literature on this subject is scarce.

The references list is suboptimal.
Response: We wonder whether the reviewer could argue more precisely why references are still considered suboptimal. This would be very helpful for us to adequately address this issue. The reference list was already enriched by including 16 additional references after the first revision of the manuscript, according to the recommendations made by Reviewer #1 and Reviewer #2. These new references are closely related to the subject, and hence appropriate and with added value for the Discussion. In fact, both reviewers were completely satisfied with the extended version of Discussion and References in the
revised manuscript. Please note that this new version includes additional references according to the new changes made, as requested.

Other comments are as in attached text.
Response: Please note that all the additional comments in relation to the text
highlighted in the pdf file attached by the reviewer have been listed below, accompanied by our responses to each comment or question raised.

Comments on the Quality of English Language
Minor editing of English is required
Response: Language revision/editing of this new version of our manuscript has been done by a native English speaker.

Introduction
“HNSCC is a highly complex heterogeneous disease…” Its better to specifically focus on laryngeal cancer. Head and Neck cancers are heterogenous, specific molecular and genomic changes is not generalized over all types of HNSCC.
Response: This has been changed in our new version of the manuscript.

“Great effort has been devoted to identify novel diagnostic or prognostic markers that reliably discriminate tumor behavior to improve patient stratification and prediction of outcome beyond current clinical and histopathological criteria. Improving HNSCC diagnosis continues to be a priority research area aimed to detect cancer at an earlier and more curable stage, which nowadays remains an unmet need. On this basis, the present study investigates the clinical significance of PIK3CA and SOX2 gene amplification in early stages of HNSCC tumorigenesis, thereby evaluating their predictive value for cancer risk assessment in 62 patients with laryngeal precancerous lesions.” This histopathological grading of laryngeal carcinoma should also be mentioned at last paragraph of introduction.
Response: To avoid confusion to the readers we consider that it should be more
adequate to maintain focus on the study subject along manuscript text by providing information related to the histopathological classification of laryngeal dysplasia. Concordantly, laryngeal dysplasia grading of our selected study cohort was precisely done according to the current WHO classification criteria. We therefore consider that histopathological grading of laryngeal carcinoma is irrelevant, as it does not apply to the present study.

Nevertheless, we have added the following further details in the last paragraph of the Introduction: “. In this regard, a new WHO classification has recently been established for laryngeal dysplasia into low-grade versus high-grade dysplasia, which attempts to overcome the lim-ited predictability of previous three-tier grading as mild, moderate and severe dysplasia [29,30]”.

Results
“Table 1. Crosstab to evaluate the correlation between PIK3CA and SOX2 gene
amplification in 62 laryngeal dysplasias”
This table is not reflective of significant genomic changes.
Response: Table 1 presents a crosstab to summarize the results quantitatively shown in Figure 1. We truly think Table 1 is informative, as it serves to easily show patient subgroups distributed according to PIK3CA and SOX2 gene amplification. However, for better clarity, we have now amended the text of Results where Table 1 is cited.

Discussion
“A new WHO classification has recently been established for laryngeal dysplasia into low-grade versus high-grade dysplasia [27,28] , which attempts to overcome the limited predictability of the previous 3-tier grading system. Noteworthy, none of these classifications significantly predicted laryngeal cancer risk in our cohort, thus reflecting the still limited value of histologic grading to accurately predict outcome.
This result emphasizes the need for additional objective and reliable markers to
improve cancer diagnosis, risk stratification and treatment decision-making.”
This article should also included the frank laryngeal carcinoma, not only dysplasia.
Early stage laryngeal ca T1-2 tumour may have significant molecular changes, in
contrast to late stage larygeal Ca T3-4 tumour.
Response: We consider that Discussion should be focused on early stages of laryngeal tumorigenesis and cancer risk assessment in laryngeal precancerous lesions (i.e. dysplasia) according to the study aim and purpose. Regarding the molecular changes that accompany laryngeal carcinomas at an early or advanced stage, this information is interesting, but beyond the topic of the study. For clarity the term ‘early’ was consistently used along the text to refer to early tumorigenesis, before malignant transformation to laryngeal carcinoma occurs.

Methods
The methodology is misplaced. It should be before Discussion segment
Response: Methods section was moved to the end of manuscript (immediately after the Discussion), according to IJMS guidelines and following IJMS manuscript template structure. This change was recommended by Reviewer #2.

Addition of the flow chart /figures summarizing the methodology will enhance the manuscript
Response: We thank the reviewer for this insightful recommendation. Please note that a graphical abstract was newly uploaded alongside our first revised version of the manuscript. Our purpose was to provide a clear and informative illustration to summarize our study, the methodology used and the most relevant findings. Nevertheless, following the reviewer’s suggestion, we have created a flow chart to specifically depict the methodological steps (new Supplementary Figure S4).

References

References list is still suboptimal!
Response: We wonder whether the reviewer could argue more precisely why references are still considered suboptimal. This would be very helpful for us to adequately address this issue. The reference list was already enriched by including 16 additional references after the first revision of the manuscript, according to the recommendations made by Reviewer #1 and Reviewer #2. These new references are closely related to the subject, and hence appropriate and with added value for the Discussion. In fact, both reviewers
were completely satisfied with the extended version of Discussion and References in the revised manuscript. Please note that this new version includes additional references according to the new changes made, as requested.